# Overhauser enhanced liquid state nuclear magnetic resonance spectroscopy in one and two dimensions

Marcel Levien [1,2,5], Luming Yang [1], Alex van der Ham [1], Maik Reinhard [1,2], Michael John [3], Armin Purea[4], Jürgen Ganz[4], Thorsten Marquardsen[4,7], Igor Tkach [1], Tomas Orlando [1,6] & Marina Bennati [1,2]

Nuclear magnetic resonance (NMR) is fundamental in the natural sciences, from chemical analysis and structural biology, to medicine and physics. Despite its enormous achievements, one of its most severe limitations is the low sensitivity, which arises from the small population difference of nuclear spin states. Methods such as dissolution dynamic nuclear polarization and parahydrogen induced hyperpolarization can enhance the NMR signal by several orders of magnitude, however, their intrinsic limitations render multidimensional hyperpolarized liquid-state NMR a challenge. Here, we report an instrumental design for 9.4 Tesla liquid-state dynamic nuclear polarization that enabled enhanced high-resolution NMR spectra in one and two-dimensions for small molecules, including drugs and metabolites. Achieved enhancements of up to two orders of magnitude translate to signal acquisition gains up to a factor of 10,000. We show that hyperpolarization can be transferred between nuclei, allowing DNP-enhanced two-dimensional $^{13}C–^{13}C$ correlation experiments at $^{13}C$ natural abundance. The enhanced sensitivity opens up perspectives for structural determination of natural products or characterization of drugs, available in small quantities. The results provide a starting point for a broader implementation of DNP in liquid-state NMR.

NMR is an indispensable tool for a wide range of applications. Compared to other analytical techniques, however, NMR spectroscopy requires relatively large amounts of material in the micro- to millimolar range concentrations, which are prohibitive for many biological samples, drugs or natural products, or conversely, necessitates excessively long experiment times or unaffordable synthetic efforts for nuclear isotope labeling[1]. The issue arises from the tiny population difference between two nuclear spin states (polarization) at room temperature

that NMR detects: at an NMR field of 9.4 Tesla, the spin polarization amounts to about 32 ppm (parts per million) for proton ($^1H$) and 8 ppm for 13-carbon ($^{13}C$), see Supplementary Equations 1. Thus, there is potential for tremendous improvement. Addressing this issue has been likely the main common challenge in magnetic resonance research over the last two decades[1–6]. Among proposed routes[7–9], NMR signals can be enhanced using a method called dynamic nuclear polarization (DNP), which provides enhancements up to

[1]Electron-Spin Resonance Spectroscopy, Max Planck Institute for Multidisciplinary Sciences, Am Fassberg 11, 37077 Göttingen, Germany. [2]Institute of Physical Chemistry, Department of Chemistry, Georg-August-University, Tammannstr. 6, 37077 Göttingen, Germany. [3]Institute of Organic and Biomolecular Chemistry, Department of Chemistry, Georg-August-University, Tammannstr. 2, 37077 Göttingen, Germany. [4]Bruker Biospin GmbH, Rudolf-Plank-Str. 23, 76275 Ettlingen, Germany. [5]Present address: Institut des Sciences et Ingénierie Chimiques, École Polytechnique Fédérale de Lausanne (EPFL), CH-1015 Lausanne, Switzerland. [6]Present address: National High Magnetic Field Laboratory, 1800 E. Paul Dirac Dr., 32310 Tallahassee, FL, USA. [7]Deceased: Thorsten Marquardsen. ✉e-mail: marina.bennati@mpinat.mpg.de

$\gamma_{\text{electron}}/\gamma_{\text{nucleus}}$, the ratio of the spins gyromagnetic ratios, which is about 660 for $^1$H and 2600 for $^{13}$C. Here, the requirement is an organic radical (the polarizing agent, PA) to be mixed with the target system and microwave (MW) irradiation at the electron spin Larmor frequency to transfer magnetization to the target nuclei of interest. Modern DNP in NMR has been established more than two decades ago in solids at cryogenic temperatures[10]. Combined with rapid sample dissolution and injection[3], DNP has opened frontiers for in vivo metabolic imaging and cancer diagnostics[11]. In solid-state NMR, DNP has permitted advances in studies of materials, surfaces and proteins[12–14]. On the other hand, a parallel trend in the most widespread field of liquid state NMR has so far not been achieved.

DNP in liquids has been known from the early days of magnetic resonance through the so-called Overhauser effect (OE-DNP)[15,16], but only recently it was discovered that DNP can deliver substantial $^{13}$C,$^{31}$P, $^{19}$F and $^1$H signal enhancements in specific molecules at high magnetic fields ($\gtrsim 3$ Tesla)[17–22]. Particularly, fast molecular collisions in the pico to sub-picosecond time scale can drive the Overhauser effect at high magnetic fields by modulating the scalar hyperfine interaction between the radical and nuclear spins on the molecule of interest[23,24]. Nevertheless, chemical systems investigated so far, including halogenated solvents, metabolites[19,25] as well as representative amino acids[26] and lipids[21,27], revealed site-specific signal enhancements. Moreover, all these studies were lacking NMR resolution as they were performed either in resonant cavities[26,28,29], deteriorating magnetic field homogeneity, or in NMR sample tubes, where high-frequency MW penetration is largely attenuated and sample heating leads to line broadening[30,31]. Here, we present the realization of a setup in which liquid DNP can be performed close to optimal and standard NMR conditions. This allows an unprecedented exploration of NMR signal enhancements for a future implementation in high-resolution NMR.

## Results and discussion
### Experimental NMR setup

Our conceived setup is schematically illustrated in Fig. 1a. It consists of an NMR magnet (9.4 Tesla, 400 MHz $^1$H resonance), a MW source and an NMR probe for DNP in liquid samples. The MW source, a frequency agile gyrotron ($\nu \approx 263.3 \pm 0.25$ GHz, $P_{\max} \lesssim 50$ W, Fig. 1a and Supplementary Note 2), is used to pump the electron spin transition of the radicals used as PA (Fig. 1c, Table 1). A commercial two-channel ($^1$H and $^{13}$C) high-resolution NMR probe (Methods) was modified to enable simultaneous MW and radio frequency (RF) irradiation (pending patent)[32]. MW is transmitted to the sample over a corrugated wave guide and a path of four mirrors (M1–M4, Fig. 1b), manufactured from Macor® ceramics and gold-coated to minimize magnetic field distortions around the sample. Importantly, M3 spreads the linearly polarized MW beam on the sample area ($\sim 20 \times 4$ mm$^2$, Supplementary Figs. 2, 3), and the amount of irradiated sample is further increased by slow spinning (20 Hz) of the sample tube around the cylinder axis.

The liquid sample is confined into a thin layer of variable thickness $d \approx 25$–75 μm, which can be tuned on the order of the MW penetration depth for specific solvents (Table 2). The sample layer is formed by confining the solution within two concentric cylindrical quartz tubes (Fig. 1d), enabling an irradiated sample volume of ~6–20 μL ("Methods"). Electromagnetic field simulations were used to calculate the value of the MW **B**-field (denoted $B_{1e}$) distribution on the sample. The simulations predicted an approximated Gaussian distribution of $B_{1e}$ along the sample axis in the z direction, with a maximum in the center of the irradiated area (Supplementary Fig. 3). The predicted $B_{1e}^{\max}$ at a MW power of 10 W is ~0.2 mT in CCl$_4$ for $d \approx 75$ μm as well as in water for $d \approx 25$ μm. Importantly, sample temperature is controlled by flow of cooled nitrogen gas past the sample. Sample temperature was calibrated using chemical shift analysis (Supplementary Fig. 4) and results showed that it is feasible to stabilize the average sample temperature

### Table 1 | g-values and 9.4 Tesla electron spin resonances of various polarizing agents

| Polarizing Agent | Solvent | $g_{iso}$ | $\Delta\nu$ (MHz) |
|---|---|---|---|
| BDPA-d$_{27}$ | toluene | 2.0025* | 0 |
| galvinoxyl | toluene | 2.0044 | 250 |
| N@C$_{60}$ | CCl$_4$ | 2.0022* | 48 |
| N@C$_{60}$ | CHCl$_3$ | 2.0022* | 48 |
| TEMPONE-$^{15}$N-d$_{16}$ | toluene | 2.0059 | 447 |
| TEMPONE-$^{15}$N-d$_{16}$ | CCl$_4$ | 2.0062 | 486 |
| TEMPONE-$^{15}$N-d$_{16}$ | CHCl$_3$ | 2.0060 | 461 |
| TEMPONE-$^{15}$N-d$_{16}$ | H$_2$O | 2.0057 | 421 |

Isotropic g-values of various PAs in selected solvents corresponding resonance frequency shifts $\Delta\nu$ with respect to that of BDPA in toluene at 9.4 T. The g-reference radicals[53] in the corresponding solvents are indicated by an asterisk. N@C$_{60}$ was calibrated in-house to a carbon fiber[52]. Precision of $g_{iso}$ is 50 ppm. Data shown in Fig. 1c and Supplementary Fig. 17. Source data are provided as a Source Data file[60].

### Table 2 | 263 GHz absorption coefficient and penetration depth

| Solvent | Sim. $\alpha$ (cm$^{-1}$) | Lit. $\alpha$ (cm$^{-1}$) | Sim. $d_{solvent}$ (μm) |
|---|---|---|---|
| H$_2$O | ~105 | ~115[55] | ~100 |
| CHCl$_3$ | ~12 | 11 – 15[56,57] | ~800 |
| CCl$_4$ | ~0.3 | 0.2[56,57] | ~3500 |

Simulated and literature values of the absorption coefficient $\alpha$ and penetration depth $d$ for H$_2$O, CHCl$_3$, and CCl$_4$ at 263 GHz[55–57]. Source data are provided as a Source Data file[60].

between 190–340 K with a gradient of $\Delta T \approx 20$–40 K during MW irradiation in various solvents, including water. The temperature gradient might still be significant and, for temperature sensitive samples, can be reduced by gating the MW or further reducing sample thickness. More technical details are reported in Methods.

Performance of the experimental setup was investigated with focus on $^{13}$C nuclei, which are crucial for structural NMR spectroscopy but rather insensitive due to the low gyromagnetic ratio and the low $^{13}$C natural abundance (1.1%). Figure 1f illustrates OE-DNP (in the following simply abbreviated with DNP) enhanced NMR spectra of model systems CHCl$_3$ in CCl$_4$ doped with the nitroxide radical TEMPONE-$^{15}$N-d$_{16}$. For pumping of the electron spin transition, MW irradiation was set on resonance with the low field line of the EPR spectrum (Fig. 1c). Other experimental details are given in the caption and in Methods. $^{13}$C signal enhancements of $\varepsilon = 120 \pm 10$ and $\varepsilon = 200 \pm 20$ were observed for CCl$_4$ and CHCl$_3$, respectively. Such enhancements correspond to a decrease in experimental time up to four orders of magnitude to obtain spectra with comparable signal-to-noise ratio (SNR), as the latter increases with the square root of acquired scans (SNR $\propto \sqrt{n}$). The trend in enhancements of the model systems ($\varepsilon$ (CCl$_4$) $< \varepsilon$ (CHCl$_3$)) is opposite as reported before at 3 and 9.4 Tesla[19,25] as we use here CHCl$_3$ diluted in CCl$_4$ instead of neat solvents. The dilution allows for a better MW penetration and saturation, consistent with the previous report at 14 Tesla[31].

With respect to previous experiments in MW cavities at the same magnetic field[25], we note an expected slight decrease in enhancements (factor of 2–3) corresponding to a saturation factor of about ~0.3 (with an error up to 25%) in the non-polar solvent CCl$_4$ ("Methods"). This corresponds to an estimated average (over the sample) $B_{1e}$ of ~0.04–0.06 mT at a gyrotron power $P_{MW} \approx 40$–50 W (Supplementary Equations 2, Supplementary Fig. 20), which is consistent with electromagnetic field calculations (Table 3). The reduced enhancement is compensated by an improvement in sample volume by a factor of ~500 (40–50 nL[25] vs 20 μL used here), which directly results in an increased NMR signal under comparable NMR detection sensitivity and frequency. A second key result is the observed line width of 7–8 Hz (full

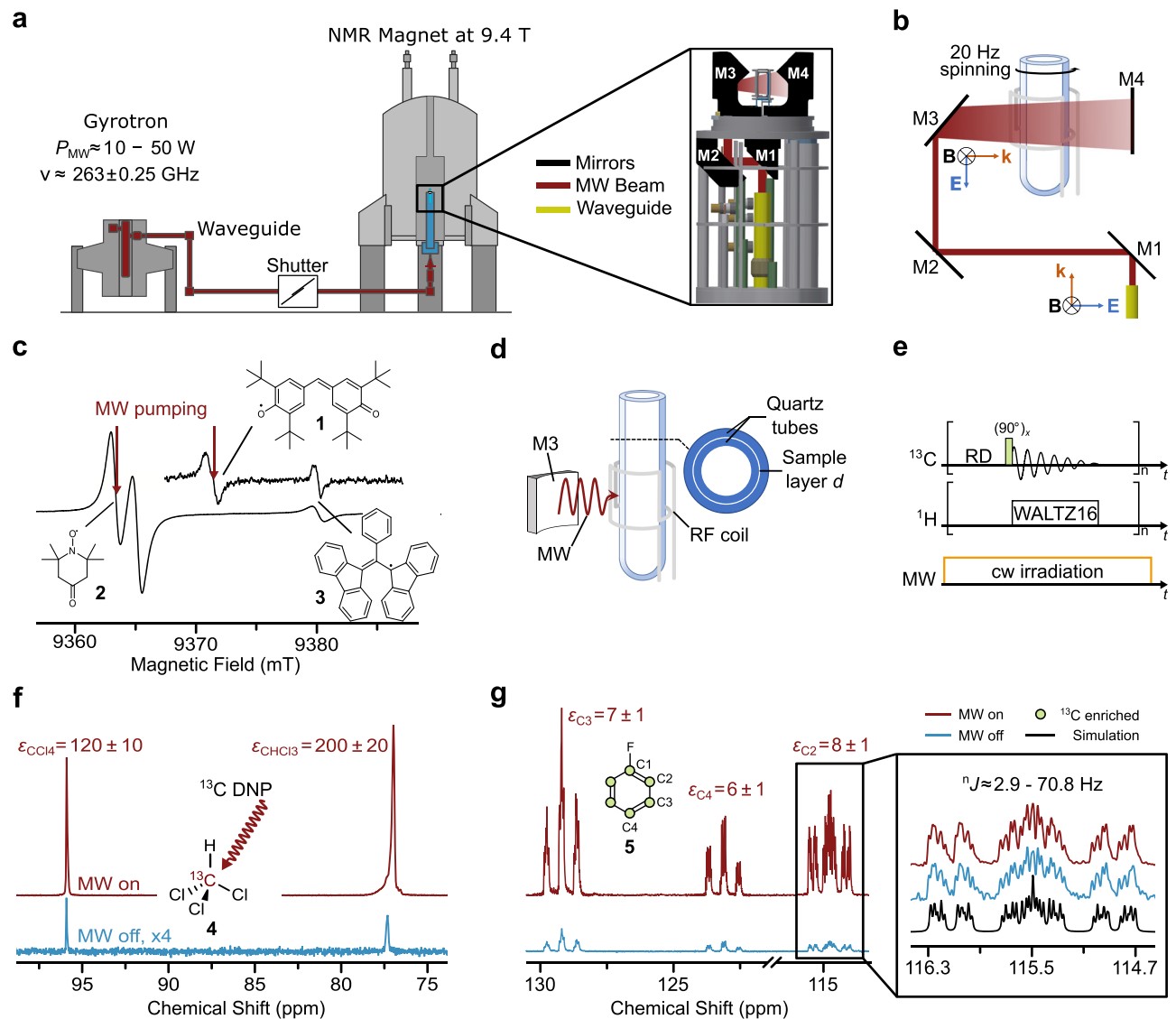

**Fig. 1 | High field liquid-state NMR spectrometer and performance. a** Schematic of the setup and zoom-in-view on the DNP probe. $\nu$ and $P_{MW}$ are MW frequency and power, respectively. **b** MW pathway and geometry of sample irradiation with mirrors (M). **k** is the vector indicating the wave propagation direction, (**E**, **B**) the electric and magnetic field vectors, respectively. **c** 263 GHz continuous wave (CW) electron paramagnetic resonance (EPR) spectra of various radicals: galvinoxyl **1**,[15]N-labeled nitroxide 4-oxo-2,2,6,6-tetramethylpiperidin-1-oxyl (TEMPONE) **2** and α,γ-bis-diphenylen-β-phenylallyl (BDPA) **3** in toluene, showing the spread of EPR resonance frequencies at 9.4 Tesla (Table 1). **d** Sketch of the sample arrangement, radio frequency (RF) coils for NMR, and MW irradiation from mirror M3 onto the sample. Concentric quartz tubes confine the sample into a thin layer of thickness $d$.

**e** Pulse sequence for 1D [13]C NMR with [1]H decoupling using the WALTZ16 pulse sequence ("Methods") and the recycle delay (RD). **f** DNP-NMR spectrum of 200 mM [13]CHCl$_3$ **4** ([13]C enriched) in CCl$_4$ doped with 10 mM of PA (TEMPONE-[15]N-d$_{16}$) (red, 4 scans) and Boltzmann equilibrium spectrum (blue, 32 scans). **g** DNP-NMR spectrum of the C$_2$-C$_4$ positions of [13]C$_6$-fluorobenzene **5** ($c \approx 500$ mM, $c$(PA) $\approx 25$ mM in CCl$_4$) recorded (red, 32 scans) and Boltzmann spectrum (blue, 32 scans). Inset shows simulation (black) of the spectrum (blue and red) according to the literature values[61]. Error in enhancements $\varepsilon$ are estimated 10–15%. Sample temperature $T$ was ~300 K, microwave power $P_{MW} \approx 40$–50 W. Figure partially reproduced from the PhD Thesis of one author[62]. Source data are provided as a Source Data file[60].

width at half height FWHH) in CCl$_4$ and 16 – 17 Hz for CHCl$_3$, which are narrower than reported so far in any high-field liquid DNP setup[25,30,31].

Another informative model system is provided by benzene and its halogenated derivatives. In Fig. 1g we display the NMR spectrum of fluorobenzene, while other enhancements are in Supplementary Figs. 5 and 23. By varying PA concentrations between 10 and 150 mM, enhancements on the aromatic [13]C vary between 6 and 20, and we are able to resolve scalar [19]F-[13]C couplings down to ~3 Hz, with [13]C line widths of ~2.3 Hz (Supplementary Fig. 7). As aromatic units form the basis of many drugs and natural products, such enhancements can be utilized for significant gain in signal acquisition time and open the door for more detailed and faster molecular investigations, as shown in the following section.

## Enhanced NMR of small molecules at natural abundance

The achieved NMR resolution permits to examine samples that contain a multitude of non-equivalent $J$-coupled nuclei. Representative enhanced spectra of the natural product trans-2-hexenyl acetate and an inhibitor of the adenyl cyclase 1 enzyme[33] are reported in Fig. 2a, b.[13]C signal enhancements on the aliphatic CH$_2$ and CH$_3$ are on the order of 3 – 14, while most carbonyl and non-protonated [13]C signals are not DNP enhanced[19,25,26]. Fig. 2c, d displays [13]C enhancements of the natural products phenylacetaldehyde and guaiazulene, as well as of the pharmaceuticals mitotane, an anticancer drug, the antiarrhythmic drug amiodarone and sodium diatrizoate (X-ray contrast agent). All individual spectra are reported in Supplementary Note 3, Supplementary Figs. 7–9. Importantly, all compounds contained [13]C at natural

**Table 3 | Comparison of the experimental and simulated $B_{1e}$ at the sample position**

| solvent | Irradiated volume (µL) | Cal. $B_{1e}^{av}$ (lin. pol.) (mT) | Cal. $B_{1e}^{av}$ (circ.pol.) (mT) | Experimental $B_{1e}^{av}$(circ. pol.) (mT) |
|---|---|---|---|---|
| $CCl_4$ | 20 | 0.033 | 0.036–0.040 | 0.045–0.065 |
| $H_2O$ | 5 | 0.027 | | – |

Simulated average (av.) $B_{1e}$ of the sample ($CCl_4$ and $H_2O$) (from Supplementary Fig. 3). Values are reported for an input power of 10 W at a frequency of 263.3 GHz. Simulated MW losses at the sample are ~4.2 dB for a 75 mm long wave guide, while experimentally ~6.5 dB were determined (Supplementary Fig. 2). Therefore, for a comparison with the experiment, the simulated $B_{1e}^{av}$ of a linearly polarized wave (lin. pol.) was scaled to the same power as used in the experiment ($P$ = 40 W). This was performed by additionally scaling the initial power ratio (40 vs 10 W) by the respective calculated and measured losses in the waveguide (4.2 vs 6.5 dB):

$$B_{1e}^{av}(\text{circ.pol.}) = \frac{1}{\sqrt{2}}\sqrt{\frac{P_{MW}^{exp}\cdot 10^{-0.65}}{P_{MW}^{sim}\cdot 10^{-0.42}}}B_{1e}^{av}(\text{lin.pol.}).$$

The factor $1/\sqrt{2}$ accounts for the fraction of linearly polarized field, which is absorbed by the nutating spin.

The obtained experimental and calculated values (last two rows of the Table) are in close agreement. Source data are provided as a Source Data file[60].

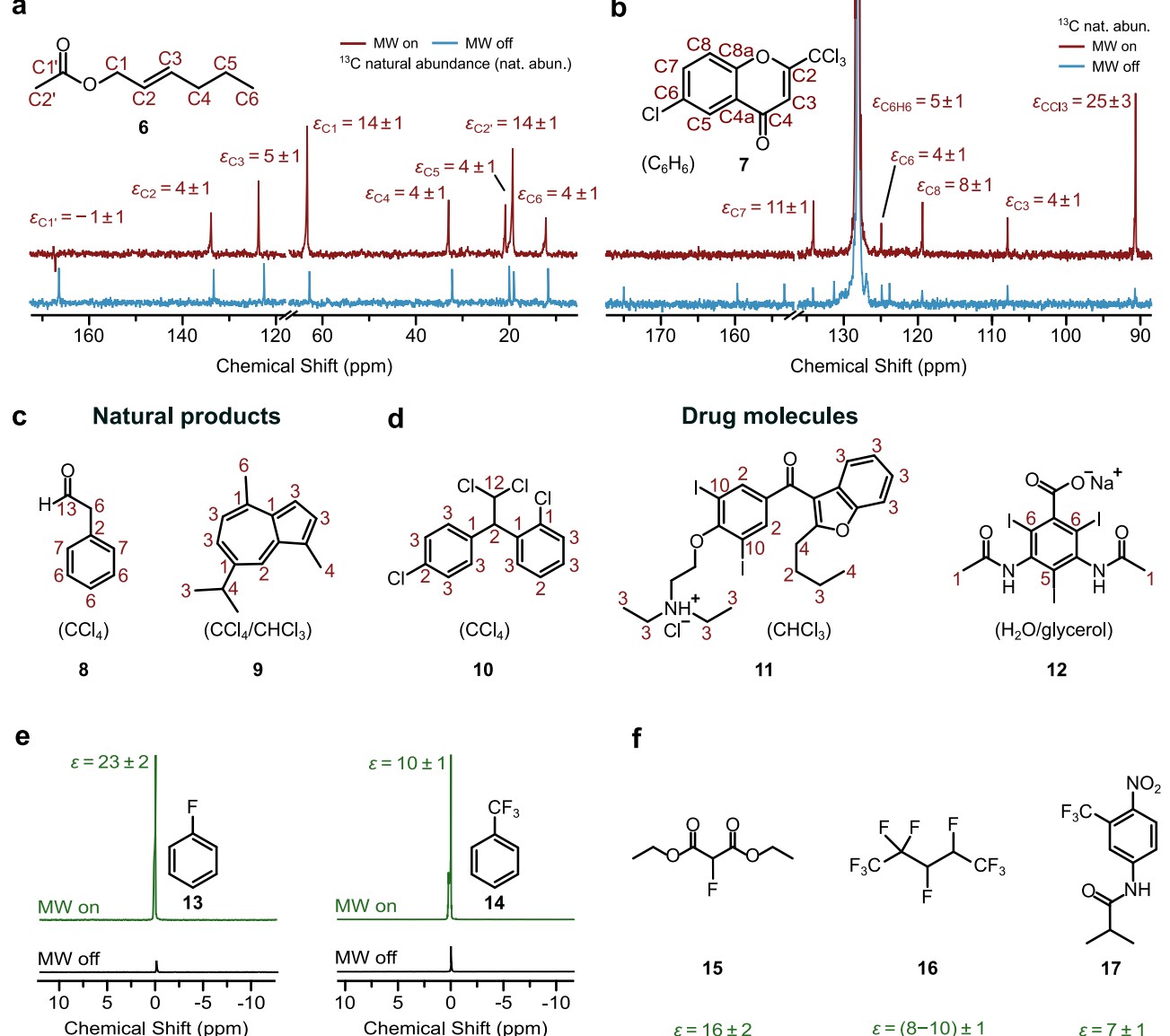

**Fig. 2 | Room temperature, natural abundance $^{13}$C as well as $^{19}$F DNP enhanced NMR spectra of small molecules. a** *E*-2-hexenyl acetate **6** ($c \approx 500$ mM, in $CCl_4$), DNP red, 128 scans, Boltzmann blue, 1024 scans. **b** NMR spectrum of an inhibitor of the adenyl cyclase 1 (AC1) **7** ($c$(PA) ≈ 100 mM, in $C_6H_6$). DNP red, 2048 scans, Boltzmann blue, 11916 scans. Spectra scaled to the same SNR. DNP enhancements $\varepsilon$ are given as red numbers. **c**, **d** DNP enhancements $\varepsilon$ of selected natural products and pharmaceutical compounds, from left to right: phenylacetaldehyde **8**, guaiazulene **9**, mitotane **10**, amiodarone HCl **11**, and sodium diatrizoate **12** (spectra in

Supplementary Figs. 8, 9). All $^{13}$C measurements were performed with inverse-gated $^1$H decoupling and with 25–100 mM TEMPONE-$^{15}$N-d$_{16}$. **e** $^{19}$F DNP NMR spectra of fluorobenzene **13** and α,α,α-trifluorotoluene **14** (DNP 4 scans, Boltzmann 4 scans) in $CCl_4$. **f** $^{19}$F DNP enhancements from left to right: diethyl fluoromalonate **15**, decafluoropentane **16**, and flutamide **17**, all samples doped with 25 mM galvinoxyl in $CCl_4$. For **17**, the solvent was $CCl_4$/DMSO. MW power adjusted to solvent polarity ($P_{MW} \approx 40$–50) and gated for aqueous samples ("Methods"). Errors of enhancements $\varepsilon$ are estimated between 10 and 25%. Source data are provided as a Source Data file[60].

abundance and solvents from a wide range of polarities (from tetra-chloromethane to chloroform and water) were employed (Fig. 2d, Table 2 and Supplementary Figs. 3, 4).

These results provide a more general picture of the scalar DNP mechanism at high magnetic fields. First, signal enhancements are observed on multiple chemical sites, however the efficiency is selective. Halogen and hydrogen atoms play a role in assisting the polarization mechanism onto a bound $^{13}$C[25,34]. Consistent with the observation on halogenated solvents ($CCl_4$ and $CHCl_3$), pronounced enhancements are observed on the $^{13}$C carbon of a $CCl_3$ group ($\varepsilon = 25 \pm 3$, Fig. 2b) as well as on the $^{13}$C in amiodarone and sodium diatrizoate bound to iodines (Fig. 2d), consistent with $\varepsilon = 57 \pm 6$ on the ipso-$^{13}$C in iodobenzene (Supplementary Fig. 5). This observation can be rationalized in the framework of the scalar DNP mechanism, according to which halogen bonds between the PA radical and the target molecule facilitate through-bond spin density transfer onto neighboring $^{13}$C atoms, as predicted by DFT calculations (Supplementary Discussions 2, Supplementary Figs. 21, 22)[24,25,34].

Large enhancements on halogenated positions are potentially useful as source of hyperpolarization, which can be distributed further using polarization transfer NMR experiments or to produce contrast in complex molecules. As a second feature, enhancements on $^{13}$C-H groups in aromatic units show some systematic trend, although the size of $\varepsilon$ slightly varies among chemical sites due to competitive binding of the PA, consistent with a previous report[26]. This appears a general feature of DNP enhancements, which, for the same chemical site, vary as a function of PA accessibility[35], local dynamics[24,36] and competition of the PA among interacting sites. To better understand this observation, we analyzed $^{13}$C DNP coupling factors at 1.2 and 9.4 Tesla (Supplementary Fig. 23, Supplementary Discussion 2). Both at 1.2 and 9.4 Tesla, $^{13}$C enhancements of the aromatic model systems are all positive, indicating a weak contribution of dipolar cross-relaxation[23]. In terms of field-dependence, this suggests that the decay of DNP efficiency toward even higher magnetic fields is shallow, as predicted by simulations of the data in Supplementary Fig. 23. This opens the perspective for application of DNP also at fields beyond 9.4 T.

Finally, we found that $^{19}$F nuclei are well suited for high-field DNP in liquids. By employing galvinoxyl radical as PA[17], (Fig. 1c) representative $^{19}$F signal enhancements of $\varepsilon \approx 10 - 25$ for fluorobenzene and trifluorotoluene (Fig. 2e) could be measured, in contrast to a negligible enhancement on the aromatic $^1$H (Supplementary Notes 3, Supplementary Fig. 10). Both molecules are important structural motifs in modern pharmaceutical drugs, such as flutamide (Fig. 2e, f and Supplementary Fig. 10), and amino acids used for in vitro as well as in-cell NMR[37]. Other examples in aliphatic systems are provided by diethyl fluoromalonate and decafluoropentane (Fig. 2f), the latter employed in industrial processes[38].

## Enhanced liquid-state two-dimensional NMR

The power of NMR spectroscopy lies in its ability to record signals from structural units of a molecule and to reveal how they are connected, which leads to structural determination. This is made possible by homo- and heteronuclear 1D and 2D NMR correlation experiments. In the following, we demonstrate the integration of liquid DNP into 2D homonuclear correlation spectroscopy on both $^{13}$C enriched and natural abundance samples. Concomitantly, we show that the feasibility of 2D experiments will allow to transfer the hyperpolarization also to less DNP sensitive sites.

In a first example, we performed DNP-enhanced 2D total correlation spectroscopy ($^{13}$C-DNP-TOCSY)[39], a method which establishes connectivities between all $^{13}$C within a given spin system. As representative sample we selected ethyl acetoacetate-1,2,3,4-$^{13}$C$_4$ (EAA) that occurs in a tautomeric equilibrium with ethyl 3-hydroxybut-2-enoate-1,2,3,4-$^{13}$C$_4$ (Fig. 3a). Whereas EAA is related to several metabolic pathways for cancer and antibiotics research[40], the pH-sensitive keto-

enol tautomerization process is relevant to several biological systems[41]. Importantly, the experiment best illustrates how the signal enhancements can be transferred from the stronger polarized $^{13}$C-H sites to adjacent, non-sensitive $^{13}$C of carbonyl groups. In the 1D pulse-acquire NMR spectrum under DNP condition, $^{13}$C enhancements $\varepsilon \approx 7-12$ are observed for the $^{13}$C of the $CH_2$ and $CH_3$ groups (Fig. 3a), while all carbonyl groups lack a DNP enhancement, similarly as in E-2-hexenyl acetate of Fig. 2a. However, in the 2D TOCSY correlation experiment the polarization is transferred from a selected source nucleus to its scalar coupled partners, i.e. from a diagonal peak to its respective cross peaks.

The 2D cross peaks under DNP conditions show greater SNR compared to those acquired under Boltzmann condition (Fig. 3c, d) and are asymmetric with respect to the diagonal peaks, due to the different initial polarization of the source nuclei after DNP enhancement. Signal enhancements of cross peaks are consistent with enhancements of the respective diagonal signals (Fig. 3c, Supplementary Figs. 11 and 12 and Supplementary Table 2). These observations were corroborated by a DNP-enhanced 2D $^{13}$C-TOCSY experiment on iodobenzene-$^{13}$C$_6$ in cyclohexane, which displayed $\varepsilon \approx 26-29$ in cross peaks correlated with the ipso-carbon (Supplementary Figs. 13 and 14, and Supplementary Table 3). Another detailed comparison of 2D cross peak intensities was performed with respect to a similar experiment performed in an unmodified 9.4 Tesla NMR instrument (no DNP, no PA, commercial NMR probe head). Data reported in Supplementary Discussions 2, Supplementary Figs. 24, 25, and Supplementary Table 11 show that the DNP TOCSY spectra of the discussed compound outperform in terms of SNR the commercial probe under comparable experimental times and absolute sample concentration.

To establish connectivities over several $^{13}$C atoms for structural determination, $^{13}$C-2D TOCSY requires $^{13}$C-enriched molecules. However, when working at natural abundance $^{13}$C, the Incredible Natural Abundance DoublE QUAntum Transfer Experiment (INADEQUATE) is one of the most attractive but challenging experiment available for NMR-based structural analyses[42]. The extension into the second dimension is critical to resolve overlapping resonances[43]. $^{13}$C-INADEQUATE (Fig. 4) maps pairwise neighboring $^{13}$C-$^{13}$C couplings through their double quantum coherences[42]. Due to the tiny 0.01% probability of encountering neighboring $^{13}$C nuclei within the same molecule, this experiment would significantly benefit from signal enhancements.

For a demonstration, we employed a natural product p-cymene, an essential oil with various pharmacological activities[44], which shows 1D $^{13}$C DNP enhancements up to $\varepsilon \sim 8$ (Fig. 4a). Notably, ~17 mg ($\sim 7.7 \cdot 10^{17}$ spins) of the analyte was employed under DNP condition to obtain the $^{13}$C-INADEQUATE spectrum after 140 hours of acquisition (Fig. 4b). Particularly, intermittent inspection of the spectra during the DNP 2D acquisition over ≲6 days (Supplementary Fig. 15) showed that DNP, NMR, and sample conditions were stable. While the Boltzmann measurement (without DNP) was not feasible in our setup, a comparison with a state-of-the-art 400 MHz $N_2$ cryo probe (Bruker, Prodigy) showed that the same experiment under similar sample condition could be acquired in 30 h (Supplementary Fig. 26). Thus, the full power of DNP might be exploited in future by combining it with cryo-probe detection.

The DNP-enhanced 2D spectrum allows assignment of all C – C connectivities, from which the molecular structure can be reconstructed. Specifically, the signals of C1 and C4, which are not observable in a 1D DNP-enhanced spectrum, are recovered by the nuclear spin polarization transfer and allows estimations of $^1J_{C1C2} \approx ^1J_{C4C3} \approx ^1J_{C2C3} \approx 57\,\text{Hz}$, $^1J_{C8C9} \approx 35\,\text{Hz}$, $^1J_{C1C7} \approx 43\,\text{Hz}$, and $^1J_{C4C8} \approx 46\,\text{Hz}$ (Fig. 4d). This is consistent with an analytical description of this experiment, according to which the double quantum coherence evolves under the product operator $\frac{1}{2}(P_1 + P_2)\sin(2\pi J_{CC}\tau)DQ_y$, where $P_1$ and $P_2$ are the

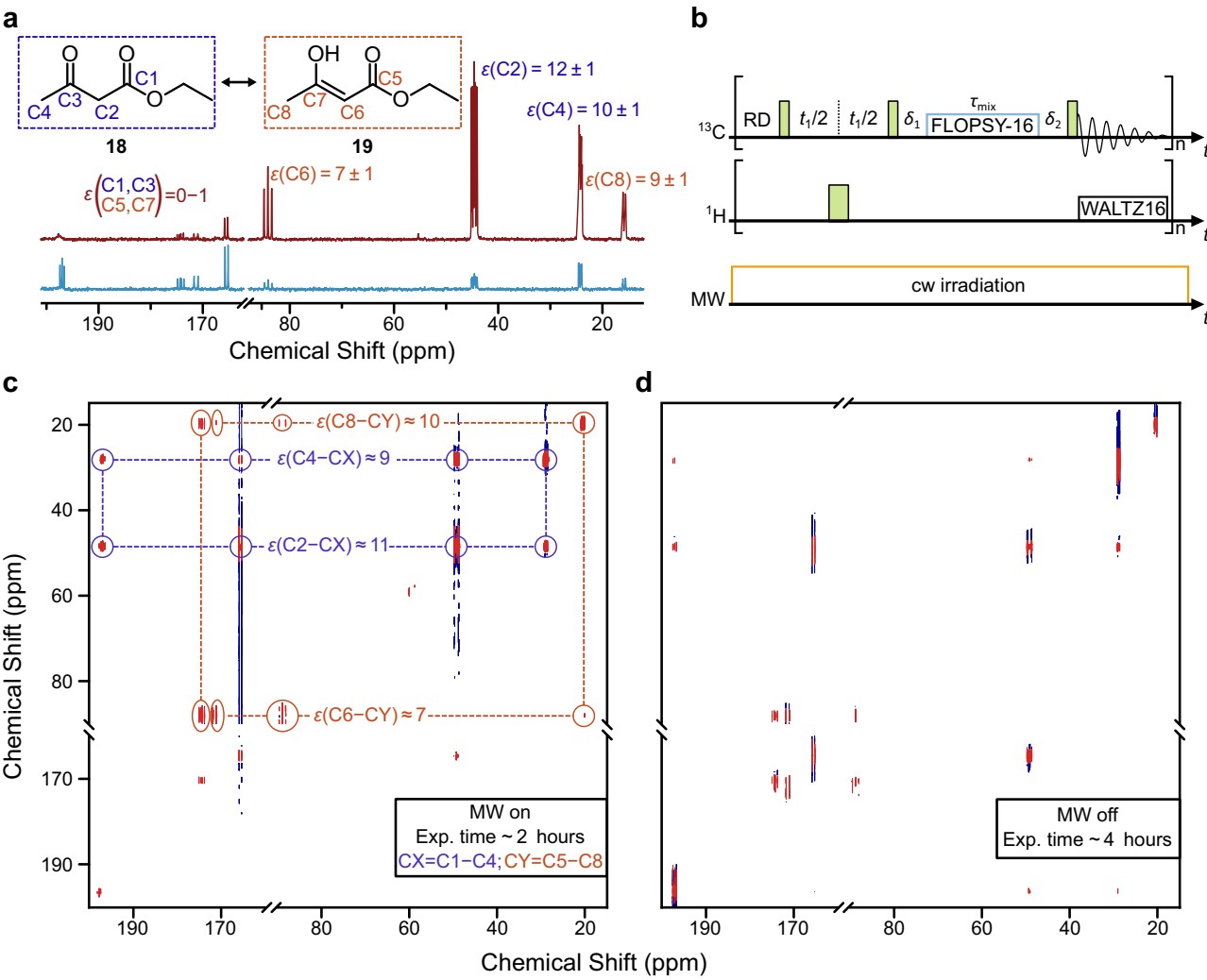

**Fig. 3 | DNP-enhanced $^{13}$C-$^{13}$C-NMR correlation experiment. a** $^{13}$C NMR spectrum of ethyl acetoacetate-1,2,3,4-$^{13}$C$_4$ **18** (carbon numbering in blue) and its tautomer **19** (carbon numbering in red), 500 mM in CCl$_4$ with 25 mM TEMPONE-$^{15}$N-d$_{16}$ under DNP (red, 32 scans) and Boltzmann (blue, 32 scans) conditions. Errors of enhancements $\varepsilon$ are estimated between 10–20%. **b** Pulse sequence for 2D $^{13}$C TOCSY with 90° and 180° pulses (green narrow and wide rectangles, respectively), heteronuclear decoupling (WALTZ16), MW irradiation, and isotropic mixing pulse

train (FLOPSY-16) ("Methods"). Time delays $\delta_1$ and $\delta_2$ were set to 2 ms and 3 ms, respectively (Methods), $t_1$ was incremented. $^{13}$C-TOCSY spectra of the same sample DNP (**c**, 8 scans) and Boltzmann (**d**, 16 scans), both with an optimized $\tau_{mix} \approx 9.4$ ms. Dashed green lines represent correlations within EAA and red lines within its tautomer. Sample temperature was -300 K, MW irradiation $P_{MW} \approx 40$ W. Experimental time is given in the inset. Source data are provided as a Source Data file[60].

polarizations of the two $^{13}$C and DQ$_y$ is the double quantum term DQ$_y = 2I_{1x}I_{2y} + 2I_{1y}I_{2x}$[45]. These observations were corroborated by a DNP-enhanced 2D $^{13}$C-INADEQUATE experiment on -7 mg of 1-fluoro-4-iodobenzene in cyclohexane (-2 · 10$^{17}$ spins), which displayed up to $\varepsilon$ - 20 in cross peaks (Supplementary Note 4, Supplementary Fig. 16). The comparison with the Boltzmann spectrum established that the enhancements of the cross peaks correspond to the average 1D enhancements of the two involved carbons.

In conclusion, an experimental design for NMR at 9.4 Tesla has allowed us to record signal-enhanced one and two-dimensional NMR spectra in liquids under conditions compatible to routine liquid NMR spectroscopy. Screening of small molecules delivered enhanced $^{13}$C and $^{19}$F signals up to two orders of magnitude, and the nuclear polarization could be spread from highly enhanced sites across networks of coupled nuclei using NMR correlation experiments. Signal acquisition under DNP did not lead to discernible sample degradation. Although direct DNP enhancements in liquids are 2–3 orders of magnitude smaller than in other hyperpolarization methods[3,4,8], in the latter cases the process is irreversible or requires specific substrates or catalysts. Instead, direct in situ DNP is performed under a steady-state condition

of hyperpolarization and thus it can be safely repeated for several purposes, i.e. signal averaging, time incrementation in multiple NMR dimensions and to transfer nuclear polarization to other sites.

Though the field of hyperpolarization has re-emerged two decades ago, it is a fact that important NMR multidimensional experiments for chemical structural determination, such as the 2D $^{13}$C-$^{13}$C INADEQUATE, could not be implemented so far in conjunction with hyperpolarization. In those other approaches, fast 2D NMR[46] has been successfully introduced to circumvent the limited life-time of hyperpolarized substrates[8,47–49], and an elegant 1D version of hyperpolarized INADEQUATE has been recently reported[43]. However, all these methods present several challenges either during the ex-situ hyperpolarization step or through substrate specificity. Therefore, direct DNP in liquids should provide an attractive avenue for routine, enhanced liquid NMR as it is compatible with standard NMR instrumentation. So far, we have demonstrated the method on small molecules and the next challenge will be an extension to macromolecules. Some avenues exist, for instance in transferring the strong hyperpolarization from solvents ($\varepsilon$($^{13}$C) $\approx 100 - 200$, Fig. 1) to macromolecules via the nuclear Overhauser effect[50], which is complicated due to the dominant

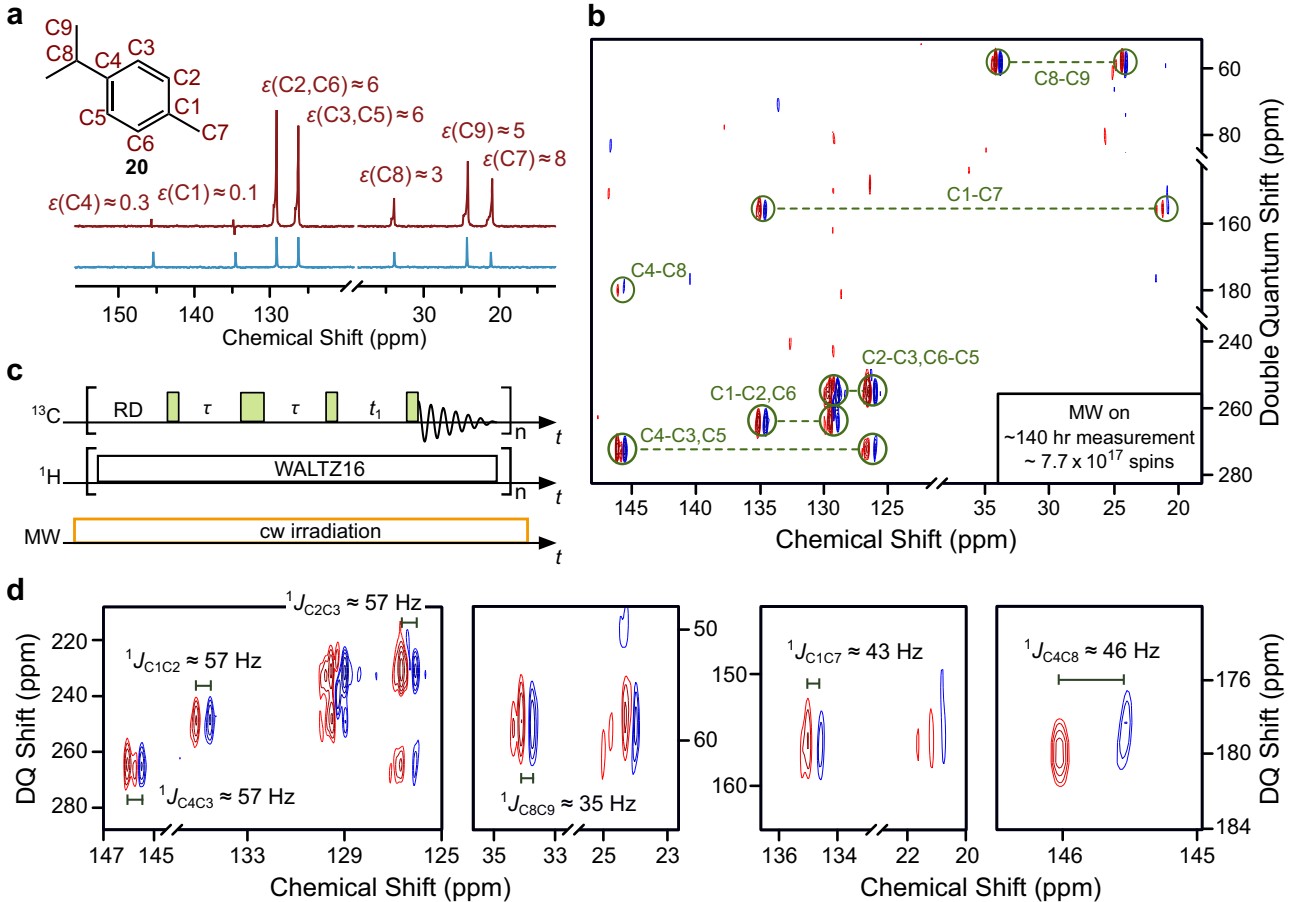

**Fig. 4 | Enhanced $^{13}$C -$^{13}$C NMR INADEQUATE experiment. a** 1D $^{13}$C NMR spectrum of neat (6.4 M) p-cymene **20** (nat. abun., carbon numbering highlighted in red) doped with 100 mM TEMPONE-$^{15}$N-d$_{16}$ under DNP (red, 64 scans) and Boltzmann (blue, 64 scans) conditions. **b** DNP-enhanced 2D $^{13}$C-INADEQUATE of ~17 mg p-cymene (nat. abun.) with 2048 × 64 points and 2560 scans. Peaks (marked by green circles) corresponding to correlated nuclei (green labels) are connected by green dashed bars. **c** Pulse sequence for DNP-enhanced 2D $^{13}$C-INADEQUATE with 90° and 180° pulses (green narrow and wide rectangles, respectively), heteronuclear decoupling (WALTZ-16) and MW irradiation ($P_{MW} \approx 40$ W) $\tau$ is set according to the scalar coupling constant $J_{CC}$, using $1/4J_{CC}$ with $J_{CC} = 45$ Hz and timing $t_1$ is incremented. RD is recycle delay. **d** Expanded views of (**b**), showing measurements of one-bond scalar-coupling constants $J_{CC}$. All measurements were performed at ~300 K. Source data are provided as a Source Data file[60].

paramagnetic relaxation. This will require new PA designs, in which the solvent/PA mixture is spatially separated from the target macro-molecule. Next developments will include further increase of sample volumes (currently our volumes are still $\lesssim$ 10-fold smaller than in a standard 5 mm NMR tube), temperature homogeneity and reduction of MW losses through materials in sample space, which would render the method compatible with low-cost MW sources. Additional improvement of sensitivity could include a combination of DNP with cryo-probe technologies and $^{1}$H detection of $^{13}$C resonances, as was already demonstrated on model systems[30,51]. Particularly the $^{1}$H dimension, which is feasible with the presented set up, is in progress and will permit to expand the capability of heteronuclear spectroscopy for structural determination of small molecules in combination with DNP. Nevertheless, the present results demonstrate liquid-state DNP in high resolution NMR.

We foresee that the simplicity of in situ DNP will boost future research, in which the full potential of liquid NMR on multiple nuclei can be exploited. Our initial results with $^{13}$C and $^{19}$F can be extended to other classes of molecules and pave the way for applications on studies of small molecules within the setting of chemical and biological analysis. Such experiments extend from characterization of drugs and drug-targets, natural products to small amounts of synthetic materials such as per- and polyfluoroalkyl substances (PFAS) in the context of environmental sciences. In conjunction with sensitivity gain, a strong

reduction of experimental time will also contribute to more sustainable analytics in academia and industry.

## Methods

### 263 GHz EPR of polarizing agents

EPR characterization ($g_{iso}$, and relaxation times $T_{1e}$ and $T_{2e}$) of the PAs in various solvents was performed with a quasi-optical EPR spectrometer (Bruker ElexSys E780) operating at magnetic fields around 9.4 Tesla. The output power of the microwave (MW) bridge is produced by an amplifier multiplier chain (AMC) and amounts to about 100 mW. Three types of resonators were used depending on the specific requirements of the experiments: a cylindrical TE$_{011}$-mode resonator for sample volumes 10 – 30 nL (Bruker BioSpin, model E9501610), a cylindrical TE$_{012}$-mode resonator with sample volumes up to ~50 nL (Bruker BioSpin, model E9501510) and a non-resonant probe for larger sample volumes (~1–4 μL, Bruker BioSpin, model E9501310). CW EPR experiments of deoxygenated samples for $T_{2e}$ measurements were acquired with the non-resonant probe. CW EPR measurements for g-factor and pulsed EPR experiments for $T_{1e}$ determination were performed with the cylindrical resonators. All experiments were recorded using quadrature detection to enable signal phase correction. Experimental parameters for CW EPR using resonator: MW power ≈ 0.2–0.5 mW (depending on solvent), modulation frequency (MF) = 100 kHz, modulation field amplitude (MA) = 0.01 – 0.05 mT,

number of scans (NS) = 1 – 10. Parameters for $T_{2e}$ measurements (Supplementary Fig. 17 and Supplementary Table 5) in the non-resonant probe: MW power ≈ 0.5–5 mW, MF = 100 kHz, MA = 0.1–0.15 mT, NS = 1. The conversion time was 81.92 ms in all CW EPR measurements.

$T_{2e}$ of the PAs were extracted from the peak-to-peak CW EPR line widths, as explained in Supplementary Discussion 1. Measurements are shown in Supplementary Fig. 17. To determine the resonance frequency of the PAs, CW EPR spectra were recorded along with $g$-standards in the same EPR capillary. $g$-standards were a carbon fiber ($g_{iso}$ = 2.00644[52]), encapsulated $^{14}N$ in $C_{60}$ (N@$C_{60}$, $g_{iso}$ = 2.0022, calibrated against the carbon fiber) or deuterated BDPA (BDPA-$d_{27}$, $g_{iso}$ = 2.0025[53]). The determined isotropic $g_{iso}$ values of the PAs in various solvents are reported in Table 1. These values were required to calculate the MW frequencies of the gyrotron for DNP pumping.

To determine the electron spin-lattice relaxation time ($T_{1e}$) at 263 GHz, Free Induction Decay (FID)-detected inversion recovery (IR) experiments were performed (Supplementary Fig. 18). Typical parameters of the pulse experiment in liquid solution were: inversion π-pulse = 160 – 200 ns, detection π/2 pulse ≈ 80 – 100 ns, τ ≈ 4 ns – 3.1 μs, dead time = 130 – 150 ns, detection bandwidth (BW) = 200 MHz, shots per point (SPP) = 50, shot repetition time (SRT) = 8 – 15 ms, number of scans (NS) = 4 – 5. $T_{1e}$ was extracted by fitting of the recovery time traces as illustrated Supplementary Fig. 18 and explained in Supplementary Discussions 1, the extracted values are reported in Supplementary Table 6.

PA concentrations were 1–50 mM and verified by comparison with standards at 9 GHz CW EPR (Bruker E500T). For 263 GHz experiments in the non-resonant probe, samples were degassed by freeze-pump-thaw cycles (3–5) in quartz tubes (I.D. = 1.6 mm, O.D. = 2 mm, Wilmad) and sealed with a flame. Samples for experiments in resonators were filled in quartz capillaries with I.D. = 0.20 mm and O.D. = 0.33 mm (Wilmad). For $T_{1e}$ experiments in resonators and under oxygen exclusion, sample stock solutions were first degassed and transferred to a glove box (MBraun-Unilab Plus, $N_2$ atmosphere, $O_2$ and $H_2O$ content ≤ 0.1 ppm). EPR capillaries were filled inside the box, sealed with sealing rubber (Critoseal®) and stored in a flask under nitrogen atmosphere prior transfer into the resonator. Sealed tubes were inserted into the resonator that was flushed with nitrogen or helium gas. Contamination of samples by oxygen was monitored over minutes up to hours as reported in Supplementary Fig. 18.

**Saturation factor and microwave field strength $B_{1e}$**
The effective saturation factor for TEMPONE-$^{15}N$-$d_{16}$ ($c$ ≈ 10 – 25 mM) in $CCl_4$ and the MW field strength $B_{1e}$ at the sample position were determined experimentally from the observed signal enhancements combined with the measured $T_{1e}$ and $T_{2e}$ relaxation times, as explained in the following.

A saturation factor $s_{eff}$ ≈ 0.3 was calculated from the Overhauser equation (eq. 2), using the signal enhancement of $CCl_4$ ($\varepsilon$ = 120, Fig. 1f) and other Overhauser parameters from the literature ($\xi$ ≈ -0.17, $f$ ≈ 0.98)[25]. A 25% error was estimated for $s_{eff}$ based on the experimental error of $\varepsilon$ and literature data.

The MW field strength $B_{1e}$ is related to $s_{eff}$ through eq. 15 (Supplementary equations 2), which is valid for a two-spin system such as TEMPONE-$^{15}N$-$d_{16}$, where one electron with $S$ = ½ couples to one $^{15}N$ nucleus with $I$ = ½. Based on the range of measured $T_{1e}$ and $T_{2e}$ (Supplementary Figs. 17–18 and Supplementary Tables 5 – 6), $s_{eff}$ was calculated and plotted as a function of $B_{1e}$ (Supplementary Fig. 20). The $B_{1e}$ value that delivered the observed $s_{eff}$ amounts to $B_{1e}$ ≈ 0.045–0.065 mT (at $P_{MW}$ ≈ 40 W). This was found consistent with the estimates from electromagnetic field simulations (Supplementary Fig. 3 and Table 3). Because $B_{1e}$ is experimentally estimated from $s_{eff}$ based on an NMR measurement, this value represents an average $B_{1e}$ over the NMR sample. The method of paramagnetic shift suppression as a function of

MW power proposed in ref. 54 is suitable for high PA concentration ($c$ ≈ 100 mM) and $s$ close to unity[54], however, it turned out difficult at our experimental conditions with low PA concentrations.

**DNP/NMR setup**
The DNP/NMR setup illustrated in Fig. 1a consists of three main components: 1) a custom-designed gyrotron as a MW source; 2) a commercial 9.4 Tesla magnet and NMR spectrometer; 3) a DNP/NMR probe. The tunable gyrotron (Bruker/CPI, 4.8 T) produces microwaves (second harmonic) at 263.3 ± 0.1 GHz at ≥ 20 W and 263.3 ± 0.25 GHz at ≤ 10 W. Specifications of the gyrotron frequency stability over time as well as the power-frequency profile as a function of hardware parameters are reported in Supplementary Note 2, Supplementary Table 1, and Supplementary Fig. 6. The linearly-polarized MW is transmitted as a free-space Gaussian beam ($TEM_{00}$-mode) to the DNP/NMR spectrometer via a corrugated waveguide (I.D. = 19.3 mm, Bruker BioSpin/CPI, mode $HE_{11}$). The waveguide consists of two segments separated by a quasi-optical bench (Bridge 12). The bench controls the power and polarizations of the MW. The bench also includes a mechanical shutter (Vincent Associates, 35 mm aperture, switch on/off time of 3 ms) and a water-cooled Teflon® absorber orthogonal to the MW pathway to absorb the reflected MW when the shutter is closed.

The NMR spectrometer consists of a commercial wide-bore 9.4 T magnet equipped with an NMR console (Bruker BioSpin Avance Neo). A liquid NMR probe (Bruker, base frequency for $^{13}C$: 100.4149 MHz; for $^{1}H$: 399.3090 MHz), with a normal coil configuration (inner heteronucleus coil, shared with $^{2}H$ channel; outer $^{1}H$ coil), was adapted here to permit MW irradiation of the sample. A corrugated waveguide (O.D. = 7.6 mm, Thomas Keating Ltd) was inserted into the central pillar of the NMR probe and coupled to the larger corrugated waveguide from the gyrotron (I.D. = 19.3 mm) via a waveguide taper (Thomas Keating Ltd). At the end of the corrugated waveguide in the probe, the 263 GHz beam is transmitted through free-space onto the sample over a combination of four mirrors and across the NMR coils (Fig. 1a, b), irradiating the sample tube from the side. The mirrors (Thomas Keating Ltd) were specifically designed to extend the electromagnetic beam waist over a sample area of about 80 $mm^2$ (Supplementary Figs. 2, 3). To allow for penetration of 263 GHz MW, the liquid sample is confined into a thin layer of thickness $d$ ≈ 25–75 μm, formed by two concentric quartz tubes. The I.D. of the outer tubes was 4.2065 ± 0.0065 mm (Wilmad Labglass 528(or 535)-PP-7QTZ) and the O.D. of the inner tube was -4.059 – 4.158 mm (Hilgenberg). This combined with the effective irradiated length of the tube (length -20 mm, Supplementary Figs. 2 and 3) results in an effective volume of ≲ 20 μL. This setup allows for tuning $d$ according to the MW absorption coefficients of various solvents and also for preserving the cylindrical arrangement of the sample for optimal shimming. Experimental characterizations of MW beam shape and polarization are given in Supplementary Fig. 2.

Sample temperature is controlled using a flow of cold nitrogen gas, which is passed through a Dewar containing liquid nitrogen, at constant flow rates (1000–1400 liters per hour). As the shortest MW irradiation time implemented in this study is on the order of seconds (on the order of $T_{1n}$), slow sample spinning at 20 Hz (a standard capability of a Bruker NMR probe) is introduced to improve homogeneity of the MW irradiation experienced by the sample, as well as to reduce the temperature gradient in the sample. Sample spinning might produce side bands, however so far, we have hardly observed spinning sidebands likely because the volume is still restricted to -20 μL and the overall SNR is in most cases too low. More details about sample characterization and temperature are given in the next sections.

**Characterization of the microwave path in the DNP probe**
Beam alignment and shape at the sample were characterized by imaging the beam spot on a liquid crystal sheet (Edmund Optics Ltd.). For this, the probe was connected to the gyrotron and the last

(M4) mirror was replaced with a camera. Liquid crystal sheet that changes color upon heating was placed at the sample position between the NMR coils, and the gyrotron was set to operate at low power $P_{MW} \approx 5 - 10$ W. The observed beam spot indicated that the beam is aligned with the sample and can be expanded over the accessible sample space between the coils, as intended by design (Supplementary Fig. 2b–d).

The MW beam polarization in the probe was examined as well. A polarizing wire grid was mounted inside the probe replacing the last mirror (M4) (Supplementary Fig. 2f). Using a vector network analyzer (VNA) as a low frequency source (9.7 GHz) that was fed into the quasi-optical bridge of the 263 GHz EPR spectrometer (Supplementary Fig. 2e), low-power 263 GHz MW irradiation was generated and directed to the probe. The reflected signal from the wire grid was down-converted (263.3 GHz to 9.7 GHz) and read out by the VNA as the $S_{11}$ (reflection) parameter. A metal plate, placed at the entrance of the probe, was used as a reference for the $S_{11}$ maximum. The observed $S_{11}$ against the grid rotation angle shows that the polarization is preserved throughout the probe and $B_{1e}$ is orthogonally oriented to the probe axis at the sample position, and thus to the external $B_0$ field in the NMR magnet (Supplementary Fig. 2g). Power losses over the entire transmission line, including mirrors, were measured to be -0.8 dB at mirror 3 (replaced by a reflector) and -6.5 dB (single path including the NMR coils, and the coil support quartz tubes, but without the sample tube assembly) at mirror 4. These values are close to electromagnetic field calculations (following section), which predict losses of -0.7 dB and ~4.6 dB at M3 and M4 (assuming a 563 mm long wave guide), respectively.

## Electromagnetic field simulations

Electromagnetic field simulations were performed to predict the $B_{1e}$ distribution over the sample using a TLM (Transmission Line Matrix) Time-Domain Solver of CST Microwave Studio® software (Dassault Systèmes). The 3D model of the probe consisted of a short, corrugated waveguide (75 mm length), four MW mirrors, two NMR coils, support and the sample-assembly quartz tubes, as well as a specific sample solvent. We used a linearly polarized Gaussian beam (TEM$_{00}$-mode) at the entrance of the corrugated waveguide with an input power of 10 W and in the frequency range of 260.0-263.6 GHz. The MW polarization at the input MW of the probe was set to reproduce the beam polarization delivered by the gyrotron.

The thickness $d$ of the sample layer was modeled as set in the experiment for the solvent type ($d \approx 75$ μm and $d \approx 25$ μm, for CCl$_4$ and H$_2$O, respectively). The values of the complex dielectric constant for quartz and CCl$_4$ at 263 GHz were extrapolated from low-frequency data (CST Microwave Studio® material library and https://cem.co.en/microwave-chemistry/solvent-choice, respectively) assuming the Debye model for frequency dependence. The resulting real parts $\epsilon_r$ and corresponding tan $\delta$ values were 3.74, 2.17 and 0.00037, 0.00041 for quartz and CCl$_4$, respectively. For H$_2$O, the high-frequency (264 GHz) experimental values $\epsilon_r = 5.36$ and tan $\delta = 1.20$, tabulated for $T = 30°$ C, were employed[55]. Extrapolation of dielectric constants by CST Microwave Studio® was validated by calculating the absorption coefficient α and comparison with literature values for propagation of a Gaussian beam ($P_{MW} = 10$ W, $\nu \approx 263.3$ GHz) through H$_2$O, CHCl$_3$, and CCl$_4$. Damping of the power density along the propagation axis of the beam was fitted to a Lambert-Beer law ($I = I_0 \exp(-\alpha t)$) to obtain α. Extracted values of α are listed in Table 2 and are consistent with literature data[55–57]. The number of mesh cells was in the range of 6.3 – 6.7×10$^7$ depending on the sample and tube properties (size and permittivity). To simplify the model, lossy metals were simulated as perfect electrical conductors (PEC). Simulations were performed using GPU-acceleration. The results from the simulations are displayed in Supplementary Fig. 3.

## Samples preparation for DNP

All solvents were purchased and used as received. Chloroform-$^{13}$C, 2,6-di-tert-butyl-α-(3,5-di-tert-butyl-4-oxo-2,5-cyclohexadien-1-ylidene)-p-tolyloxy (galvinoxyl), 4-Oxo-2,2,6,6-tetramethylpiperidine-d$_{16}$,1-$^{15}$N-1-oxyl (TEMPONE-$^{15}$N-d$_{16}$), ethyl acetoacetate (EAA), ethyl acetoacetate-1,2,3,4-$^{13}$C$_4$, chlorobenzene-$^{13}$C$_6$, bromobenzene-$^{13}$C$_6$, iodobenzene-$^{13}$C$_6$, phenylacetaldehyde, ST034307, α,α,α-trifluorotoluene, diethyl fluoromalonate, decafluoropentane, and flutamide were obtained from Sigma-Aldrich. Amiodarone hydrochloride, sodium diatrizoate and, mitotane were obtained from Fisher Scientific. Trans-2-hexenyl acetate was obtained from TCI chemicals. 1-fluoro-4-iodobenzene was obtained from Fluka. Fluorobenzene was obtained from EGA. Fluorobenzene-$^{13}$C$_6$ was obtained from Cambridge Isotope Laboratories. For all DNP experiments, commercial 5.0 mm O.D. quartz tubes were used (Wilmad Labglass). Solutions of polarizing agent (-35–75 μL) were degassed within the NMR tube by at least five freeze-pump-thaw cycles. Samples were then transferred to a nitrogen filled glovebox (MBraun) where a quartz insert ~4.1 mm O.D. (Hilgenberg) was inserted and the tubes were capped using airtight caps made in-house. EPR spectra were then recorded at 9 GHz (X-band, Bruker Elexsys E500T, Elexsys high sensitivity probe), line intensities and width were compared to standards to verify the PA concentration and check deoxygenation. Typical $^{13}$C and $^{19}$F DNP samples were prepared with $c$(PA) ≈ 10 – 100 mM and $c$(target) ≈ 200–500 mM. The p-cymene samples were prepared by dissolving the PA in neat p-cymene. In the case of flutamide, we used a target concentration of ~10 mM and ~330 mM of DMSO to account for the poor solubility of the compound. Radical concentrations were optimized for the best compromise of signal enhancement and NMR line width of the target molecule.

## Overhauser DNP experiments

Before MW irradiation, the sample temperature was stabilized at reduced temperature (210–270 K) under a flow of cold nitrogen gas. The effective sample temperature was verified based on the chemical shift difference of the solvent signals, as determined in a separate experiment, using a sample containing no radical, but otherwise identical composition. Depending on the solvent, the temperature dependence of the chemical shift difference was monitored over a range of 70–90 K without MW irradiation and compared to the chemical shift difference during CW MW irradiation at different MW power (Supplementary Fig. 4a–b) or varying MW irradiation time (Supplementary Fig. 4c–e). From the comparison, temperature settings for the specific solvents were obtained. For DNP we applied MW irradiation on-resonant with the PA low field line (Fig. 1c). By adjusting the gun current, the cavity temperature, the collector current and the cathode voltage of the gyrotron (Supplementary Note 2), the MW frequency was first set at the resonance condition estimated from the isotropic g-factor of the PA and then kept fixed (Table 1).

Magnetic field homogeneity was optimized by shimming, and the lock-field (± 1.1 mT) swept to record a DNP enhancement field profile. The lock-field was then set to the maximum of the profile, corresponding to the low-field EPR line of the PA. All NMR spectra were recorded without field locking. The NMR probe was tuned and matched to nuclei under investigation. All experiments were at least duplicated under similar experimental conditions. From this, the error in enhancement was estimated to be ~10%. In cases where the signal-to-noise ratio of the Boltzmann spectrum was poor, an increased uncertainty of ~15% was assumed.

Pulse-acquire experiments were performed with 10.5 μs (~41 W, 25.5 kHz) and 14.7 μs (~21 W, 17.01 kHz) high power 90° pulses for $^{13}$C and $^{1}$H, respectively, whereas $^{1}$H decoupling was performed with WALTZ16 using a nominal 90 μs pulse (~0.58 W, 2.78 kHz). If not noted otherwise, 1D $^{13}$C DNP experiments were performed with $^{1}$H decoupling during acquisition but without $^{1}$H pre-saturation.

[19]F pulse acquire experiments were carried out by retuning the [1]H NMR coil to the resonance frequency of [19]F (~375 MHz) with a 17 μs high power pulse (~21 W, 14.71 kHz). All other experimental details were similar to [13]C 1D DNP experiments.

DNP enhancements were calculated as the ratio of the absolute integral of the signal in the absence (Boltzmann signal) and presence (DNP) of MW irradiation. Spectra were processed with the same parameters (exponential line broadening, zero-filling, zero- and first-order phase correction). When the number of scans between these two experiments was not the same, the absolute integrals were multiplied/divided accordingly. For samples, in which the Boltzmann signal was too weak to be recorded in the DNP-tube, the Boltzmann spectrum was collected in a 5 mm regular NMR tube, and intensities were scaled by the volumetric ratio.

The recycle delay (RD) was set to ~1.2–5 times $T_{1n}$ and no dependence of the enhancement on RD was observed. For measurements with $H_2O$ as a solvent, a longer RD of max. 30 s was used to account for sample heating.

### Two-dimensional NMR/DNP experiments

DNP spectra of all 2D correlation measurements were collected under CW MW irradiation at output power of $P_{MW} \approx 40$ W. The [13]C TOCSY experiments were performed a slightly modified version of a standard pulse sequence provided in the TopSpin software package (dipsi2ph), which was modified by including broad band decoupling on [1]H during acquisition, and with incorporation of an additional 180° pulse on [1]H during the $t_1$ period (Fig. 3b). FLOPSY-16 and DIPSI-2 spin locks, both with $\omega_1/2\pi = 10$ kHz and zero-quantum filters of $\delta_1 = 2$ ms and $\delta_2 = 3$ ms[58] are used to account for the different chemical shift dispersion of EAA (~17.7 kHz) and iodobenzene-[13]C$_6$ (~4.5 kHz). Carrier frequencies of the spin-lock were set at the center of the chemical shift ranges.

[13]C-TOCSY spectra of EAA were collected on a $CCl_4$ solution containing 500 mM target molecule and 25 mM TEMPONE-[15]N-d$_{16}$. Spectra were collected with $\tau_{mix} = 9.4$ ms spin-lock, 8 (OE-DNP) and 16 (Boltzmann) scans, 16 dummy scans, 6 s recovery delay, 32768 × 128 complex points, zero-filled to two times the number of data points, and processed with a shifted sine-bell filtering in both dimensions. The [13]C-TOCSY spectra of iodobenzene-[13]C$_6$ were collected on a cyclohexane solution containing 500 mM target molecule and 25 mM TEMPONE-[15]N-d$_{16}$. Spectra were collected with $\tau_{mix} = 34.5$ ms spin-lock, DIPSI-2 spin lock, 8 scans, 16 dummy scans, 6 s recovery delay, 4096 × 256 points, zero-filled to four times the number of data points, and processed with a shifted sine-bell filtering in both dimensions. Corresponding 1D spectra of EAA were collected with 32 scans, 32768 points, 30 s recovery delay, and processed with 2 Hz exponential broadening. 1D spectra of iodobenzene-[13]C$_6$ were collected with 8 scans, 16384 points, 30 s recovery delay, and processed with 2 Hz exponential broadening.

The DNP-enhanced [13]C-INADEQUATE spectrum of neat p-cymene (6.4 M) with 100 mM TEMPONE-[15]N-d$_{16}$ as PA was obtained as a summation of five spectra, each recorded with 512 scans and 16 dummy scans. Each spectrum was collected with 3 s recovery delay, 2048 × 64 points, zero-filled to four times the number of data points, and processed with a shifted sine-bell filtering in both dimensions. Inter pulse delay for the double quantum evolution was set to $\tau = 5.55$ ms and optimized for observing [13]C homonuclear scalar-coupling with $J_{CC} = 45$ Hz. To keep track of the sample condition, 1D pulse-acquire measurements were performed after each of the five 2D spectra (Supplementary Note 4, Supplementary Fig. 15).

The DNP-enhanced [13]C-INADEQUATE spectrum of 1-fluoro-4-iodobenzene (natural [13]C abundance) was collected on a cyclohexane solution containing 1.5 M target molecule and 25 mM TEMPONE-[15]N-d$_{16}$ and with 288 scans. The reference Boltzmann spectrum was collected with the same target and PA concentration as the DNP measurement in a regular NMR tube with total volume of ~0.6 mL and with 128 scans. In both cases, spectra were recorded with 16 dummy scans, 6 s recovery delay, 2048 × 128 points, zero-filled to four times the number of data points, and processed with a shifted sine-bell filtering in both dimensions. Inter-pulse delay for the double quantum evolution was set to $\tau = 4.5$ ms and optimized for observing [13]C homonuclear scalar-coupling with $J_{CC} = 55$ Hz. WALTZ-16 and WALTZ-64 broadband heteronuclear decoupling[59] ($\omega_1/2\pi = 2.78$ kHz) were implemented for the [13]C-TOCSY and [13]C-INADEQUATE experiments, respectively.

## Data availability

Source data are provided with this paper[60]. The Source Data provided in this study have been deposited at the Göttinger Research Online Data Base under accession link [https://doi.org/10.25625/AQY3SI].

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

## Acknowledgements

We gratefully acknowledge the European Research Council (ERC Advanced Grant 101020262 BIO-enMR (M.B.) and the Max Planck Society for the financial support. M.L. thanks the Göttinger Graduate School GGNB for a PhD fellowship, L. Y. the Alexander von Humboldt foundation for a postdoctoral fellowship, and T.O. the DFG Project number 455993474. M.R. and M.B. acknowledge the DFG through the BENCh Research Training Group – 389479699/GRK2455. We thank also Leonard Bröker for initial experiments with $^{19}F$, Sandra and Gareth Eaton for their comments on $T_{1e}$ and $T_{2e}$ nitroxide relaxation, Christian Griesinger and Matthias Ernst for critically reading this manuscript,.

## Author contributions

M.B. supervised the project. M.B., T.O., I.T., M.L. conceived the idea and together with A.P. and T.M. designed the experimental setup. A.P. and J.G. constructed the DNP-NMR probe. M.L., L.Y., T.O. and I.T. characterized the setup. I.T. performed electromagnetic field simulations. M.L., L.Y., A.H., M.R. performed 1D NMR experiments, L.Y. and M.L., A.H. and M.J. performed 2D NMR experiments and analyzed the NMR data. I.T. and M.L. performed and analyzed 263 GHz EPR experiments. All authors contributed to the discussion of the project. M.B., M.L. and L.Y. wrote the manuscript with input from all co-authors except for T.M. who sadly passed away before the writing phase.

## Funding

## Competing interests

J. Ganz (J.G.), A. Purea (A.P.) and T. Marquardsen (T.M.) are employees of Bruker Biospin. Bruker Biospin was not involved in any data collection, analysis and decision to publish. The international patent application PCT WO 2024/115698 A1, indicated in Ref. 32, was filed by Bruker BioSpin GmbH togehter with Max Planck Society (Max-Planck-Gesellschaft zur Förderung der Wissenschaften) and Thomas Keating Ltd., with the following inventors: T. Marquardsen, M. Bennati, I. Tkach, L. Marcel, T. Orlando, L. Yang, A. Leveasley, R. Wylde. The PCT application was published on June 6th, 2024 and contains detailed design and dimensions of the DNP probe as illustrated in Fig. 1a, b and Supplementary Fig. 3. The patent contains representative examples of the probe performance as illustrated in Fig. 1–f, Supplementary Fig. 7a and Supplementary Fig. 13. The remaining authors declare no competing interests.
