## [Peer Review File · Nature Communications]

Overhauser enhanced liquid state nuclear magnetic resonance spectroscopy in one and two dimensionsREVIEWER COMMENTS

Reviewer #1 (Remarks to the Author):

This manuscript describes an impressive paradigm shift in the sensitivity of liquid-state NMR experiments, based on Overhauser liquid-state Dynamic Nuclear Polarization (DNP). The low sensitivity of NMR is known to be its major limitation, and the last two decades have witnessed tremendous efforts to circumvent it. While DNP is now commonly used in solid-state NMR, it is not yet the case in liquid-state NMR. Dissolution-DNP and para-hydrogen induced spectroscopy allow tremendous sensitivity improvements but suffer from fundamental limitations that prevent their general use. Overhauser-DNP, which enables the implementation of DNP in a liquid-state setting, is much more general but until now, it has been limited to very small sample volumes at high magnetic field, or to experiments at very low magnetic fields.

Here, the authors show that an impressive improvement in the experimental design allows the acquisition of hyperpolarized liquid-state NMR spectra at high field (9.4 T), with a reasonable sample volume (a few μL). This makes the Overhauser-DNP approach quite general and able to tremendously enhance the sensitivity of detection for a variety of small molecules. Furthermore, the authors show that such hyperpolarization can be transferred between nuclei, allowing the acquisition of a variety of 2D experiments at natural ^{13}C abundance, with a highly enhanced sensitivity. These elements are very novel and original, to the best of my knowledge.

The manuscript is extremely well written, and the findings are supported by very clear descriptions, data analysis and complementary investigations (including DFT calculations to better understand the polarization transfer mechanisms. I am convinced that the results reported here are a game changer for liquid-state NMR, and have the potential to revolutionize the field in the coming years. Therefore, I strongly support publication of this very nice paper in Nature Communications. I only have a few minor remarks:

-the experimental setting requires slow spinning of the sample. The authors might want to comment on potential associated drawbacks. Did they notice any spinning sidebands, as it is the case in conventional liquid-state NMR (where sample spinning is not recommended for routine 2D experiments)?

-the authors mention a 20-40 K temperature gradient during the experiments, which seems quite huge. Could they comment on this drawback, in particular how it can affect the spectrum quality for molecules with temperature-dependant chemical shifts? Are there molecules for which the acquisition of 2D spectra would be impossible due to this temperature gradient?

-the authors report an impressive 140 h INADEQUATE experiment with their setting, but they also say that Boltzmann measurement was not feasible with their setup. However, with more than 10 mg of sample, a conventional INADEQUATE spectrum should be easily recorded at high-field in a few hours, maybe with a cryoprobe (see D. Uhrin, *Annu. Rep. NMR Spectrosc.* 2010, Vol. 70, doi:10.1016/S0066-4103(10)70004-1). It would be fair to report such a spectrum as supplementary information.

-the authors declare no conflict of interest. I think this is not acceptable since there are authors from Bruker, who obviously have such conflict of interest. This should be clearly stated.

Reviewer #2 (Remarks to the Author):

The manuscript by Bennati and coworkers presents a milestone in magnetic resonance spectroscopy. They have invented a smart new hardware design for room-temperature DNP solution NMR, which resolved many long-lasting key technical challenges in this exciting field. Empowered by their thin-layer setup in a cylindrical NMR tube and excellent microwave engineering, a range of 1D and 2D ^{13}C NMR spectroscopy can be now conducted on reasonable amount of

organic compounds of natural ^{13}C abundance. The authors deployed their new technique to explore a wide range of chemicals, including drugs and natural products, demonstrating the great potential in many important applications. They have also identified several key features of ODNP-enhanced ^{13}C NMR spectroscopy, e.g. the chemical selectivity and the asymmetry of homonuclear ^{13}C - ^{13}C 2D NMR spectroscopy. The manuscript is well structured and easy to read. All the figures are well prepared. This manuscript matches the scope and readership of Nat. Commun.. It will bring significant impacts in several relevant fields.

Despite the undoubtedly high value of this work, several minor points have to be addressed:

1. I understand the joy triggered by the impressive results on n.b. compound as shown in Fig. 4. In particular, this demo highlights the stability of the whole system. However, the authors must be aware that 140 h acquisition for a ^{13}C - ^{13}C INADEQUATE spectrum on 17 mg of such a simple (and really small) organic compound is still away from the reality in organic chemistry studies. This remaining gap has to be addressed clearly in order to prevent misled interpretations by non-expert users.

2. The authors should also address the need of ^1H dimension in future, as well as potential challenges and opportunities under the current DNP. In fact, the utilization of ^{13}C hyperpolarization and more sensitive ^1H detection has been recently shown by several groups. In real life of organic chemistry and med chem, the structure determination without ^1H dimension would be, if not impossible, rather tedious.

3. The transverse ^{13}C relaxation times in the absence and presence of radicals should be provided, as these data will be necessary for evaluating the sensitivity loss in complex 2D experiments.

4. There are quite a large body of discussions on differential ODNP enhancements on different chemicals and chemical groups. The authors should rather present a summary and a statistical presentation of these results obtained on all studied molecules.

5. The DNP enhancement on CCl_4 (120 ± 10) is significantly lower than that obtained on CHCl_3 (200 ± 20). This trend is opposite to the previous observations (930 on CCl_4 vs. 550 on CHCl_3 @ 3 T , 430 ± 50 on CCl_4 vs. 320 ± 60 on CHCl_3 @ 9.4 T). The authors should explain the inconsistency, in particular at the same B_0 field.

Reviewer 1:

This manuscript describes an impressive paradigm shift in the sensitivity of liquid-state NMR experiments, based on Overhauser liquid-state Dynamic Nuclear Polarization (DNP). The low sensitivity of NMR is known to be its major limitation, and the last two decades have witnessed tremendous efforts to circumvent it. While DNP is now commonly used in solid-state NMR, it is not yet the case in liquid-state NMR. Dissolution-DNP and para-hydrogen induced spectroscopy allow tremendous sensitivity improvements but suffer from fundamental limitations that prevent their general use. Overhauser-DNP, which enables the implementation of DNP in a liquid-state setting, is much more general but until now, it has been limited to very small sample volumes at high magnetic field, or to experiments at very low magnetic fields.

Here, the authors show that an impressive improvement in the experimental design allows the acquisition of hyperpolarized liquid-state NMR spectra at high field (9.4 T), with a reasonable sample volume (a few μL). This makes the Overhauser-DNP approach quite general and able to tremendously enhance the sensitivity of detection for a variety of small molecules. Furthermore, the authors show that such hyperpolarization can be transferred between nuclei, allowing the acquisition of a variety of 2D experiments at natural ^{13}C abundance, with a highly enhanced sensitivity. These elements are very novel and original, to the best of my knowledge.

The manuscript is extremely well written, and the findings are supported by very clear descriptions, data analysis and complementary investigations (including DFT calculations to better understand the polarization transfer mechanisms. I am convinced that the results reported here are a game changer for liquid-state NMR, and have the potential to revolutionize the field in the coming years. Therefore, I strongly support publication of this very nice paper in Nature Communications. I only have a few minor remarks:

We are very grateful to the reviewer for these positive comments.

1. the experimental setting requires slow spinning of the sample. The authors might want to comment on potential associated drawbacks. Did they notice any spinning sidebands, as it is the case in conventional liquid-state NMR (where sample spinning is not recommended for routine 2D experiments)?

This is a valuable question. In the current setup, we have hardly observed spinning sidebands. Under DNP conditions, the sample volume is still restricted to a few \$\mu\text{L}\$ and therefore the overall S/N is likely too low to observe spinning sidebands. Additionally, the

NMR lines are, in some cases, broadened at the foot/bottom due mainly to the temperature gradient, which likely prevents from resolving spinning sidebands. In future developments, where larger sample volumes and larger enhancements will be targeted, suited phase cycling might be considered if side bands are observed.

In order to clarify this, we have added following sentence in the methods section p. 16: *Sample spinning might produce side bands, however so far, we have hardly observed spinning sidebands likely because the volume is still restricted to ~20 μ L and the overall S/N is in most cases too low.*

2. the authors mention a 20-40 K temperature gradient during the experiments, which seems quite huge. Could they comment on this drawback, in particular how it can affect the spectrum quality for molecules with temperature-dependent chemical shifts? Are there molecules for which the acquisition of 2D spectra would be impossible due to this temperature gradient?

The reported temperature gradients are still significant (Fig. S4), however they vary from sample to sample and, in temperature-sensitive samples, can be reduced by gating the MW or reducing the sample layer thickness. In the latter case, also the DNP enhancements will be reduced. While in ^{13}C spectra, the temperature gradient is, in most cases, not reflected in line shapes (see Fig. 1 g), ^{19}F NMR spectra are more sensitive due to their temperature-dependent chemical shifts (Fig.2 main text and Fig.S10). At this stage, we cannot provide a generalized answer to the question as to whether 2D spectra for some particular molecules would be impossible, many more data are required. However, we can state that -so far- the temperature gradient was not an obstacle in the presented investigations of small molecules, except when using water as solvent, where enhancements were reduced. Following sentence was added on page 5:

The temperature gradient might still be significant and, for temperature sensitive samples, can be reduced by gating the MW or further reducing sample thickness.

3. the authors report an impressive 140 h INADEQUATE experiment with their setting, but they also say that Boltzmann measurement was not feasible with their setup. However, with more than 10 mg of sample, a conventional INADEQUATE spectrum should be easily recorded at high-field in a few hours, maybe with a cryoprobe (see D. Uhrin, Annu. Rep. NMR Spectrosc. 2010, Vol. 70, doi:10.1016/S0066-4103(10)70004-1). It would be fair to report such a spectrum as supplementary information.

Our setup presents a significant improvement as compared to previously reported liquid-state OE-DNP setups. However, there is still a gap in sensitivity as compared to state-of-the-art commercial liquid state NMR. To avoid confusion and to quantify this gap, following the suggestion of the reviewer, we now report in Fig. S26 a 2D INADEQUATE spectrum

of the same sample as Fig. 4 recorded on a commercial liquid-state NMR instrument with a nitrogen cryo-probe (Bruker, Prodigy). As expected, the cryo-probe provides a better S/N (a factor 3 - 5 in average over the peaks) than the DNP set up as latter shows a factor of two worse sensitivity for ^{13}C as compared to a standard probe (Table S9). We reformulated the main text on page 11 accordingly:

While the Boltzmann measurement (without DNP) was not feasible in our setup, a comparison with a state-of-the-art 400 MHz N_2 cryo probe (Bruker, Prodigy) shows a factor of about 3 - 5 better sensitivity in the latter (Fig. S26). Thus, the full power of OE-DNP might be exploited in future by combining it with cryo-probe detection.

4. the authors declare no conflict of interest. I think this is not acceptable since there are authors from Bruker, who obviously have such conflict of interest. This should be clearly stated.

We have filed a disclosure of potential conflicts of interest according to the Nature Research journal policy. The declaration of competing interests on page 24 has been changed accordingly.

Reviewer 2:

The manuscript by Bennati and coworkers presents a milestone in magnetic resonance spectroscopy. They have invented a smart new hardware design for room-temperature DNP solution NMR, which resolved many long-lasting key technical challenges in this exciting field. Empowered by their thin-layer setup in a cylindric NMR tube and excellent microwave engineering, a range of 1D and 2D ^{13}C NMR spectroscopy can be now conducted on reasonable amount of organic compounds of natural ^{13}C abundance. The authors deployed their new technique to explore a wide range of chemicals, including drugs and natural products, demonstrating the great potential in many important applications. They have also identified several key features of ODNP-enhanced ^{13}C NMR spectroscopy, e.g. the chemical selectivity and the asymmetry of homonuclear ^{13}C - ^{13}C 2D NMR spectroscopy. The manuscript is well structured and easy to read. All the figures are well prepared. This manuscript matches the scope and readership of Nat. Commun.. It will bring significant impacts in several relevant fields. Despite the undoubtably high value of this work, several minor points have to be addressed:

We thank the reviewer for the positive comments.

1. I understand the joy triggered by the impressive results on n.b. compound as shown in Fig. 4. In particular, this demo highlights the stability of the whole system. However, the authors must be aware that 140 h acquisition for a ^{13}C - ^{13}C INADEQUATE spectrum on 17 mg of such a simple (and really small) organic compound is still away from the reality in organic chemistry studies. This remaining gap has to be addressed clearly in order to prevent misled interpretations by non-expert users.

This comment is very similar to comment #3 of reviewer 1. We thank both reviewers for pointing this out, as the previously missed comparison is now very meaningful. A comparative inadequate spectrum under same sample condition was recorded with a commercial N_2 cryoprobe and the data added as Fig.S26. The better performance of the cryo-probe versus the DNP probe (factor of about 3 - 5 in sensitivity) indicates that the potential of DNP can and should be exploited in future by combining DNP with cryoprobe technology.

2. The authors should also address the need of ^1H dimension in future, as well as potential challenges and opportunities under the current DNP. In fact, the utilization of ^{13}C hyperpolarization and more sensitive ^1H detection has been recently shown by several groups. In real life of organic chemistry and med chem, the structure determination without ^1H dimension would be, if not impossible, rather tedious.

Our liquid-state DNP probe is indeed equipped with two radio frequency channels (^1H and ^{13}C) allowing for many different ^1H NMR experiments including ^1H decoupling or detection. Almost all presented ^{13}C spectra were acquired using ^1H decoupling. Furthermore, for ^{19}F DNP experiments, the ^1H channel was tuned to the resonance frequency of ^{19}F . ^1H detection using polarization transfer schemes such as reversed (^{13}C to ^1H) Inensitive Nuclei Enhanced by Polarization Transfer (INEPT) has already been reported in the literature (ref 30 and 50 main text). Detection in the ^1H dimension is therefore accessible and 2D experiments such as 2D Heteronuclear Correlation Spectroscopy are in progress. This point has been stressed in the conclusions:

Particularly the ^1H dimension, which is feasible with the presented set up, is in progress and will permit to expand the capability of heteronuclear spectroscopy for structural determination of small molecules in combination with DNP.

3. The transverse ^{13}C relaxation times in the absence and presence of radicals should be provided, as these data will be necessary for evaluating the sensitivity loss in complex 2D experiments.

We report now the nuclear transverse relaxation data of p-cymene in figure S27.

4. There are quite a large body of discussions on differential ODNP enhancements on different chemicals and chemical groups. The authors should rather present a summary and a statistical presentation of these results obtained on all studied molecules.

We understand the reviewer's point in summarizing our experimental observations on OE-DNP enhancements of specific chemical environments. However, we note that the enhancement does not solely depend on the direct chemical environment. Aspects that are also highly relevant for the OE-DNP effect include accessibility of the target molecule/radical (Levien et al. *PCCP*, **2021**), local dynamics (Levien et al. *JPCL*, **2020** and Orlando et al. *JMRO*, **2022**), and also competition of different interaction sites within the target molecule. For this reason, as because this is the very first study reporting this kind of data, we would like to refrain from a statistical evaluation of the enhancements as a function of chemical sites. We highly appreciate the comment and will devote further attention to the topic. We account for this by inserting a sentence on page 8:

This appears a general feature of DNP enhancements, which, for the same chemical site, vary as a function of PA accessibility, local dynamics and competition of the PA among interacting sites.

We also added Levien et al. *PCCP*, **2021** and Levien et al. *JPCL*, **2020** to the reference list.

5. The DNP enhancement on CCl₄ (120 +/- 10) is significantly lower than that obtained on CHCl₃ (200 +/- 20). This trend is opposite to the previous observations (930 on CCl₄ vs. 550 on CHCl₃ @ 3 T, 430 +/- 50 on CCl₄ vs. 320 +/- 60 on CHCl₃ @9.4 T). The authors should explain the inconsistency, in particular at the same B₀ field.

In our current study, we report $\varepsilon(\text{CCl}_4) < \varepsilon(\text{CHCl}_3)$ using CCl₄ as the solvent and CHCl₃ as the target molecule. A similar trend for CCl₄ and CHCl₃ is observed at 14 T when using pentane-d₁₂ as a solvent ($\varepsilon(\text{CCl}_4) = 21$, $\varepsilon(\text{CHCl}_3) = 70$, Dubroca et al. *PCCP*, **2019**). On the other hand, we previously reported $\varepsilon(\text{CCl}_4) > \varepsilon(\text{CHCl}_3)$ at 9.4 T using pure solvents (Orlando et al. *ACIE*, **2019**). We attribute this discrepancy to reduced MW absorption using non-polar solvents as compared to earlier measurements in pure solvents. This is confirmed by our measurements at 9.4 T using pure CCl₄ and CHCl₃, respectively. In this case, we also observe a larger enhancement for CCl₄ ($\varepsilon(\text{CCl}_4) = 135$, $\varepsilon(\text{CHCl}_3) = 90$). The data are reported in the Figure 1 below. Following sentence was added on page 5:

The trend in enhancements of the model systems ($\varepsilon(\text{CCl}_4) < \varepsilon(\text{CHCl}_3)$) is opposite as reported before at 3 and 9.4 Tesla as we use here CHCl₃ diluted in CCl₄ instead of neat solvents. This dilution allows for a better MW penetration and saturation, consistent with the previous report at 14 Tesla.

Figure 1: (a) Signal enhancement of CCl_4 doped with 10 mM TEMPONE- $^{15}\text{N-d}_{16}$ with a MW power of $P \approx 40 - 50$ W. (b) Signal enhancement of CHCl_3 doped with 25 mM TEMPONE- $^{15}\text{N-d}_{16}$. Due to increased MW absorption of CHCl_3 , the maximum enhancement was observed at a MW power of $P_{\text{MW}} \approx 30$ W and with a thinner sample layer as compared to measurements in CCl_4 .

REVIEWERS' COMMENTS

Reviewer #1 (Remarks to the Author):

The authors have seriously taken my comments into consideration, as well as those from other reviewers. I recommend acceptance of the manuscript as it stands.

Reviewer #2 (Remarks to the Author):

The authors have addressed properly and explicitly all the questions. This work is highly important and should be published.